# Ultrastructural Characterization of Human Gingival Fibroblasts in 3D Culture

**DOI:** 10.3390/cells11223647

**Published:** 2022-11-17

**Authors:** Sandra Liliana Alfonso García, Laura Marcela Mira Uribe, Susana Castaño López, Monica Tatiana Parada-Sanchez, David Arboleda-Toro

**Affiliations:** 1Department of Oral Health, Faculty of Dentistry, Universidad Nacional de Colombia, Bogotá 111321, Colombia; 2Department of Integrated Basic Studies, Faculty of Dentistry, Universidad de Antioquia, Medellín 050010, Colombia; 3School Bioscience, Faculty of Science, Universidad Nacional de Colombia, Medellín 050034, Colombia

**Keywords:** 3D culture, gingival fibroblasts, amorphous calcium phosphate, periodontics

## Abstract

Cell spheroids are applied in various fields of research, such as the fabrication of three-dimensional artificial tissues in vitro, disease modeling, stem cell research, regenerative therapy, and biotechnology. A preclinical 3D culture model of primary human gingival fibroblasts free of external factors and/or chemical inducers is presented herein. The ultrastructure of the spheroids was characterized to establish a cellular model for the study of periodontal tissue regeneration. The liquid overlay technique was used with agarose to generate spheroids. Fibroblasts in 2D culture and cell spheroids were characterized by immunofluorescence, and cell spheroids were characterized by optical and scanning electron microscopy, energy-dispersive X-ray spectroscopy, backscattered electrons, and Fourier transform infrared spectroscopy. Ostegenic related genes were analyzed by RT-qPCR. Gingival fibroblasts formed spheroids spontaneously and showed amorphous calcium phosphate nanoparticle deposits on their surface. The results suggest that human gingival fibroblasts have an intrinsic potential to generate a mineralized niche in 3D culture.

## 1. Introduction

Oral mucosae are an appropriate cell source for regenerative medicine, as they are easily accessed through minimally invasive procedures without generating esthetic problems [1]. Human gingival fibroblasts (hGFs) are cells with desirable phenotypic characteristics for regenerative strategies of soft and mineralized craniofacial tissues because they can undergo a variety of alternative destinations, such as myofibroblasts, pericytes, endothelial cells, and induced pluripotent stem cells (iPSC), in response to extrinsic mechanical or chemical factors [2]. Increased cellular plasticity and proliferation rate in favor of hGFs have been also suggested when comparing their gene expression profile with lining mucosa fibroblasts [3]. Current research indicates that fibroblast-mediated regeneration relies on the activation of embryonic or neonatal gene expression programs across different organs. Furthermore, the reprogramming or rejuvenation of differentiated cells such as fibroblasts into a fetal or neonatal stage in three-dimensional (3D) culture conditions generates a biomechanical response in the fibroblasts that induces their reprogramming into stem-cell-like cells. These partially reprogrammed fibroblasts not only exhibit stem-cell-like characteristics but also retain their differentiation states to some extent [4,5].

Gingival tissues have been reported as an adequate cell source to increase the cellularity of a periodontal defect; hGFs represent a promising option, with regenerative potential for periodontal tissues. A recent randomized clinical trial showed that grafting a scaffold made of β-calcium triphosphate (β-TCP) with autologous hGFs covered by a collagen membrane in periodontal defects induced a significant reduction in vertical pocket depth and reduced attachment loss. In addition, radiographic evaluation revealed statistically increased bone gain compared to the control group treated with β-TCP followed by a coverage of the defect with a collagen membrane [6]. Similarly, in another clinical trial, autologous hGFs were cultured on a collagen matrix, and a connective tissue graft was placed under a coronally advanced flap for the treatment of single and multiple gingival recessions. This approach produced a greater increase in the width of the keratinized tissue than in the control group. Therefore, hGFs used as a novel tissue-engineering and living cell-based therapy were demonstrated to be efficient in gingival recession treatment and proved to be a reliable and successful treatment [7]. Additionally, in an animal model, autologous gingival fibroblasts were used as a type of seeding cells in a sandwich tissue-engineered complex to repair periodontal defects. By using this approach, the researchers achieved an ideal periodontal reconstruction in a short time [8]. However, the intrinsic capacity of hGFs in periodontal regeneration is not widely known.

Developing in vitro models for the study of cell biology and cell physiology is of considerable importance to the fields of disease modeling, cancer research, drug discovery, toxicity testing, stem cell research, regenerative therapy, and biotechnology. Although two-dimensional (2D) cell culture has been a well-accepted method to culture cells in vitro for more than a century and has significantly contributed to the understanding of cell behavior, it does not provide a vivo-like environment; therefore, physical cues, cell–cell and cell–matrix communication, and the interplay of different cell types are difficult to reproduce. In contrast, the 3D culture technique can overcome these difficulties and even provide mechanical cell reprogramming without genetic or biochemical interventions [9]. Such 3D culture systems can be classified into scaffold-free or anchorage-independent 3D culture systems, anchorage-dependent or scaffold-based 3D culture systems, and specialized 3D culture platforms. Scaffold-based techniques include liquid overlay, which is one of the most explored methodologies, owing to its low cost, easy handling, and spontaneous spheroid formation [10].

Cells within spheroids are exposed to physical interactions that closely reflect their behavior within 3D native tissue. They exhibit enhanced cell viability, regular protein secretion, cell differentiation ability, stable morphology, and polarization, as well as increased proliferative activity and physiological metabolic function. These properties may be the result of the 3D spherical structure of the spheroids, which allows for ample chemical/and cell–ECM interactions [11].

The objective of this research is the ultrastructural characterization of spheroids derived from hGFs free of external factors and chemical inducers as a safe and clinically relevant alternative for therapeutic purposes, as well as the development of a cellular model for periodontal tissue research and translational dentistry.

## 2. Materials and Methods

### 2.1. Tissue Collection

The study was carried out according to the principles of the Declaration of Helsinki for biomedical research and was approved by the Medical Ethics Committee of the Clinic Noel Foundation (24 August 2018) and the Ethics Committee of the Faculty of Dentistry of the University of Antioquia-Medellin, Colombia (record N° 01, 5 February 2016). Written and oral informed consent were obtained from all donors.

The study included 7 subjects with nonsyndromic cleft lip and/or cleft palate (NS CL ± P), comprising 5 males and 2 females, with ages ranging from 3.5 to 10.5 months, who were consulting the multidisciplinary team of the Noel Clinic Foundation of Medellin, Colombia. A tissue sample was taken from the area of the palatal wrinkles close to the cleft palate. A 2 mm diameter surgical punch was taken by a plastic surgeon. The size of the biopsy was 2 mm in length and 2–3 mm in thickness, including all connective tissue. Inclusion criteria included a clinical diagnosis of NS CL ± P with no presentation of any other type of associated clinical abnormality that indicated the presence of a syndrome, as well as no presentation of previous corrective surgery. Tissue specimens were transported to the laboratory at 4 °C (iced container) in falcon tubes with 2 mL high-glucose Dulbecco’s modified Eagle medium (DMEM) (Thermo Fisher Scientific, Waltham, MA, USA) supplemented with 5% fetal bovine serum (FBS) (Sigma Aldrich, Saint Louis, MO, USA) and 10% antibiotics (100 U/mL penicillin G, 10,000 µg/mL streptomycin) (Thermo Fisher Scientific, Waltham, MA, USA).

### 2.2. Primary Cell Culture

Primary cultures of hGFs cells were established using the explant culture technique. Tissue specimens were washed with phosphate-buffered saline (PBS) (Thermo Fisher Scientific, Waltham, MA, USA), cut into pieces of approximately 0.5 mm in size, placed and seeded in T25 cm^2^ culture flasks for adherent cells, and cultured in DMEM supplemented with 10% FBS and 1% antibiotics (penicillin/streptomycin) at 37 °C in a humidified atmosphere with 5% CO_2_. The medium was changed 3 times a week in a proportion of 1:1. The outgrowth of fibroblasts from surgical fragments was observed after 25 days. When fibroblasts reached 80% confluency, fragments were removed, and cells were detached with 0.05% trypsin and 5 mM EDTA (Thermo Fisher Scientific, Waltham, MA, USA) for 4 min. Cells were subcultured and expanded in T75 cm^2^ tissue culture flasks with a seeding density of 3 × 10^3^ cells/cm^2^ in complete growth medium. Cells from passages 3 to 6 were used for the following experiments.

### 2.3. Spheroid Formation and Culture

Spheroids were generated from 2D hGFs using 96-well, flat-bottomed cell culture plates coated with agarose. A 1% solution of agarose (VWR Life Science, Radnor, PA, USA) was prepared in PBS and sterilized by autoclaving. A total of 30 µL of the solution was added per well. Agarose-coated dishes were left at room temperature for 1 h and exposed to UV light before cell seeding.

For spheroid induction, cells were subcultured as mentioned above. Trypsin was neutralized with PBS supplemented with 10% FBS. Then, cells were centrifuged at 223 g for 5 min to obtain a cell pellet, which was resuspended in a fresh culture medium. To generate spheroids, a cell suspension containing 5 × 10^3^ cells in 150 µL of complete culture medium (DMEM) was used. A total of 60 spheroids were seeded for each of the samples. Wells at the edges of the plate were filled with PBS to account for evaporation of the medium. Approximately 30 µL of fresh medium was added the day after seeding, and the medium was subsequently changed every 3 days.

Growth and morphology of the 3D spheroids were monitored every 24 h. An average of 20 of 60 spheroids were randomly selected from each of the samples to perform the morphological analysis. Optical images were carried out using an inverted Zeiss Axiovert 200 (Carl Zeiss, Göttingen, Germany) equipped with an AXIOCAM ERC5S camera (Micron CMOS Far sensor, 2.2 × 2.2 µm pixel size, 8 bit/pixel); (Carl Zeiss, Jena, Germany). Open-source ImageJ version number 1.53h free software (Wayne Rasband and contributors National Institute of Health, Bethesda, MD, USA) was used to measure diameter, area, perimeter, solidity, and roundness. The behavior of the spheroids was evaluated through the time of cultivation.

All data were expressed as the mean ± standard deviation (SD). A one-way ANOVA test, Kruskal–Wallis, and Levene tests were used to test the statistical significance with R free software (Lucent Technologies, Murray Hill, NJ, USA). *p* values < 0.05 were considered statistically significant.

### 2.4. Immunostaining of 2D and 3D Cultures

Human gingival fibroblasts seeded on glass coverslips at a density of 30,000 cells per well in a 24-well culture plate were incubated for 48 h. Spheroids were harvested from cultures 3 and 4 days after initial induction. Whereas cells in the monolayer were washed with cold PBS and fixed with 4% paraformaldehyde for 20 min, whole spheroids were fixed for 45 min. Cells in 2D and 3D cultures were then permeabilized using 0.25% Triton X–100 (Thermo Scientific, Waltham, MA, USA) for 10 min. Unspecific binding was blocked using 1% BSA/0.1% Tween 20 in PBS (Thermo Scientific, Waltham, MA, USA) at room temperature for 30 min.

The populations of hGFs cultured in the monolayer and in spheroids were characterized by immunofluorescence staining using the following primary antibodies: anti-human vimentin (Vim) (1:1000) (Abcam, Cambridge, UK), anti-human collagen type I alpha 2 chain (Col1A2) (Proteintech, Rosemont, IL, USA) (1:1000), anti- human α-smooth muscle actin (αSMA) (1:50) (Proteintech, Rosemont, IL, USA), anti-human integrin βeta1 (Itgβ1) (1:50) (Proteintech, Rosemont, IL, USA), anti-human platelet-derived growth factor receptor beta (PDGFRβ) (1:20) (GeneTex, Irvine, CA, USA), anti-human stromal interacting molecule 1 (STIM1) (1:50) (Abcam, Cambridge, UK), and anti-human Stro-1 (Stro-1) (Abcam, Cambridge, UK). All primary antibodies were incubated overnight at 4 °C. After incubating the cells with the secondary antibodies, Alexa Fluor 594-conjugated goat anti-mouse (1:1000) (Abcam, Cambridge, UK), or Alexa Fluor 488-conjugated goat anti-rabbit (1:1000) (Abcam, Cambridge, UK), nuclei were stained as required with 4′,6-diamidino-2-phenylindol (DAPI, Sigma Aldrich, Saint Louis, MO, USA) for 1 h. To mount the stained cells on the slide, Vectashield mounting medium was dispensed on the slide, and the coverslip was placed over the mounting medium to disperse medium over the entire section. The immunostaining of both fibroblast monolayers and whole spheroids was visualized on a fluorescence microscope (Nikon Eclipse 80i, Tokyo, Japan), and the images were obtained using an OptixCam Summit K2 camera (Microscope LLC, Roanoke, VA, USA) and processed with Fiji/ImageJ version number 1.53c (plus) (University of Wisconsin; Madison, WI, USA) free software.

### 2.5. Scanning Electron Microscopy, Energy-Dispersive X-ray Spectroscopy, and Backscattered Electrons

Spheroids were fixed overnight in 2.5% glutaraldehyde prepared in Sorensen’s phosphate-buffered solution (pH 7.2). Spheroids were then washed in PBS, fully dehydrated in a graded series of ethanol solutions, and dried. The samples were fixed on graphite tape, thinly coated with gold (Au) (Denton vacuum desk IV equipment, Moorestown, NJ, USA), and analyzed by scanning electron microscopy (SEM, Jeol JSM 6490 LV, JEOL, München, Germany) at the university research headquarters, University of Antioquia, Colombia. Subsequently, some areas of the spheroid surfaces were analyzed by energy-dispersive X-ray spectroscopy (EDS). Areas of interest of the spheroids were selected according to the topographic analysis previously carried out on the SEM for chemical characterization of nanoparticles present on the surface of the spheroids by means of EDS. The concentrations in weight expressed as a percentage of the chemical elements present in the areas of interest of the spheroids were identified in three zones of the spheroids of all samples. Finally, backscattered electrons (EB) analysis was performed on specific areas of the spheroids according to the previous EDS analysis. The same samples were used for all analyses.

### 2.6. Fourier Transform Infrared Spectroscopy

Fourier transform infrared spectroscopy was performed to identify the basic structural components, as well as the conformation of the most relevant biological molecules on the surface of hGFs spheroids. To this end, selected spheroids were cultured at a density of 5 × 10^3^ cells for 21 days in two 96-well plates. Then, spheroids were isolated and fixed in 2.5% glutaraldehyde in Sorensen’s phosphate buffer (pH 7.2) for 1 h at room temperature. After three washes in PBS, they were transferred to a slide and dried in a dry air oven at 50 °C for 24 h. The analyses were performed in an IRAffinity-1s infrared spectrometer (Shimadzu, Kyoto, Japan). Samples were suspended in spectroscopic-grade potassium bromide (KBr) pellets and analyzed by transmittance in the wavenumber range of 500 to 4000 cm^−1^ with a resolution of 4 cm^−1^ and 16 scans per sample.

### 2.7. RNA Extraction, cDNA Synthesis, and Real-Time Quantitative PCR (RT-qPCR)

The expression of mechanotransduction marker genes and osteogenic activity-related genes was analyzed by RT-qPCR. Total RNA was extracted from 3D spheroids on day 4 and from 2D cultures when they reached 70–80% confluence using a RNeasy mini kit (Qiagen, Hilden, Germany) according to the manufacturer’s instructions. The purity and concentration of RNA were evaluated at 260 nm and 260 nm/280 nm ratios using an a Nanodrop 2000 spectrophotometer (Thermo Fisher Scientific, Waltham, MA, USA). Complementary DNA (cDNA) was synthesized from 350 ng of extracted RNA using the an Applied Biosystems high-capacity kit (Thermo Fisher Scientific, Waltham, MA, USA). The reaction was performed as recommended by the manufacturer, with a first step at 25 °C for 10 min, a second step at 37 °C for 120 min, a third step of heat inactivation at 85 °C for 5 min, and a final step at 4 °C.

According to the results obtained from the calibration curves, the concentration of 0.2 ng/mL of cDNA was used for the RT-qPCR analysis. The reaction was performed in a 20 µL final volume using a QuantiTect SYBR Green kit (Qiagen, Hilden, Germany) using 10 µL of 2X master mix buffer and 1.2 µL each of forward and reverse primers (0.3 µM concentration). The volume was completed with RNase-free water, and each reaction was performed in triplicate. Following the manufacturer’s instructions, the PCR assay was carried out under the following conditions: hot start at 95 °C for 15 min, denaturation at 94 °C for 15 s, alignment for 30 s at the specific temperature for each set of primers, and a last elongation step at 72 °C for 30 s and 40 cycles. The RT-qPCR reaction was performed using a RotorGen Q 5-Plex HRM thermocycler (Qiagen, Hilden, Germany). Rotor-Gene Q series software version number 2.3.1 (Qiagen, Hilden, Germany) was used to analyze the Ct values and melting curves of each sample. Expression results of target genes were normalized against the average Ct values of glyceraldehyde 3-phosphate dehydrogenase (*GADPH*) and 60S ribosomal protein L27 (*RPL27*) for each of the 2D and 3D culture conditions. The double-delta Ct method (2^–∆∆ct^) was used to calculate the expression ratio of each sample in the two culture conditions. The list of RT-qPCR primers used is shown in Table 1. Primers were designed or verified using the Primer BLASTS program version number 2.2.25 (NCBI, Bethesda, MD, USA) and were synthesized by Macrogen, Inc (Seoul, South Korea).

## 3. Results

### 3.1. Human Gingival Fibroblasts Self-Assemble Spheroids with the Liquid Overlay Technique

Tridimensional hGFs spheroids were successfully generated by the liquid overlay technique. In general, during the spontaneous formation of spheroids, cells go through three stages—aggregation, compaction, and spheroid formation—without the need for additional stimuli (Figure 1A). Gingival fibroblast spheroids were aggregated and achieved the compaction stage approximately 48 h after cell seeding and reached the consolidation stage on day 4 of culture.

To investigate changes in size after spheroid formation, parameters of diameter, area, perimeter, roundness, and solidity were evaluated for 10 days. In general terms, a decrease in diameter, area, and perimeter was observed over the time course. After 48 h, the fibroblasts aggregated and formed a single spheroid with an average diameter of 35.890 ± 72.83 µm in most of the wells. The diameter decreased as the cultivation time increased, reaching a diameter of 26.144 ± 41.5 µm on day 10 of culture (Figure 1B), corresponding to an overall rate of decrease in the spheroidal diameter of 28.5% during this incubation time. As a consequence, the area also decreased from 6671 ± 22 mm^2^ on day 2 of culture to 4056 ± 1219 mm^2^ on day 10 of culture, demonstrating spheroid compaction. The diameter and spheroidal area also reached similar dimensions on day 4 of culture for all samples without statistically significant differences (*p* value < 0.05) (Figure 1B). The average diameter observed on day 4 for all samples was 290.6 ± 37.4 µm, and the area was 5220 ± 1719 mm^2^. These measurements correspond to medium-sized spheroids.

Perimeter, an indicator of surface roughness, significantly decreased on day 3 and remained constant during the remaining culture period for most patients. In contrast, the roundness and solidity increased with the culture time. From the third day, there was an increase in roundness (sphericity), which was slight but constant until the end of the culture, reaching values between 0.82 and 0.91% (0.86% on average), indicating an ellipsoidal tendency of the structure due to cell detachment or, in the case of hGFs, due to the budding of one or more small secondary spheroids. Likewise, solidity increased from day 3 and remained stable until day 10 of culture, with values between 0.89 and 0.94, indicating a high cell concentration (Figure 1B).

### 3.2. Human Gingival Fibroblasts Exhibited Mesenchymal Cell Markers

Gingival fibroblasts in 2D culture were positive for a subset of markers: Vim, Col1A2, PDGFRβ, Itgβ1, and αSMA. In contrast, they were negative for STIM1 and Stro-1 markers, as shown in Figure 2A.

The spheroids continued expressing the mesenchymal markers (Vim, Col1A2, PDGFRβ, Itgβ1, and αSMA). However, unlike the monolayer culture, the spheroids expressed STIM1, a protein involved in controlling the entry of positively charged calcium ions into the cells when the levels of the ions were low, specifically through calcium-release activated calcium (CRAC) channels (Figure 2B). By sensing luminal Ca^2+^ concentration, STIM1 protein, in turn, is responsible for relaying the signal of Ca^2+^ store depletion to pore-forming Orai1 proteins in the plasma membrane. A direct interaction of STIM1 and Orai1 allows for the re-entry of Ca^2+^ from the extracellular space. Therefore, the expression of STIM1 suggests that 3D culture of hGFs induced cell response to changes in the mechanical environment of the niche mediated by store-operated calcium entry (SOCE)/CRAC activation.

### 3.3. Human Gingival Fibroblast Spheroids Form a Highly Cohesive Structure with Deposits of Mineral Nanoparticles on Their Surface

During the first days of culture, spheroids exhibited rounded cells on their surface (Figure 3A). However, it was also evident that hGFs spheroids became highly cohesive and showed tight intercellular connections, forming a flat surface with increased culture time. The strong interaction between cells resulted in a compact spheroid structure. Additionally, the presence of multiple pores was observed on the spheroidal surface, regardless of the cultivation time (Figure 3B,C). As an important finding, the presence of multiple spherical microparticles was identified, with average diameters between 0.5 and 0.9 µm in different zones of the surface of the spheroids. (Figure 3D,E).

The EDS analysis identified oxygen [O], carbon [C], calcium [Ca], and phosphorus [P] elements, mainly in the areas where the nanoparticles were found, that is, on the surface of the spheroids from day 2 of culture until day 21 of culture. In addition, the weight ratio (expressed as percentage) was variable for each ion, depending on the zone analyzed. Nanoparticle conglomerates were observed on the surface of spheroids after 3 days (Figure 4A) and 21 days (Figure 4C) of culture. The EDS analyses are presented with the corresponding weight percentages of the elements identified in Figure 4B for the spheroid formed after 3 days of culture and in Figure 4D for the spheroid formed after 21 days of culture. In addition, the SEM image of a spheroid after 9 days of culture is shown with a higher magnification (Figure 4E). The qualitative and quantitative analysis by EDS of this spheroid detected the following ions in descending weight percentage: O (47.38%), Ca (26.71%), Si (16.54), and Fe (9.38) (Figure 4F). Moreover, the same deposit of nanoparticles was observed on the surface of the spheroid with a lower magnification by SEM (Figure 4G) and EB analysis, showing that the increased dispersion of electrons ensured an image with increased brightness in the same area where the nanoparticles were located, indicating the formation of a structure with heavier chemical elements than in the rest of the spheroidal surface (Figure 4H).

In general, the weight ratio (expressed in percentage) was variable for each of the ions, depending on the zone analyzed and the time of cultivation. For example, Ca percentages between 15.20 and 4.90% were observed, whereas the P content was found to be between 1.70 and 2.60%. Similar behavior was observed for C and O.

### 3.4. Three-Dimensional Culture Induces the Synthesis of Amorphous Calcium Phosphate by hGFs

FTIR analysis of hGFs spheroids cultured for 21 days under standard conditions identified bands related to water bound to the sample (peak 1) at 3444 cm^−1^, as well as lipids possibly associated with the cell membrane in the range of 3000–2700 cm^–1^ (peaks 2, 3). Additionally, it confirmed the presence of vibrational bands corresponding to phosphates [PO_4_^3−^] (peaks 11 and 12) and carbonates [CO_3_^2−^] (peak 7) of inorganic nature. Furthermore, FTIR identified the amide groups corresponding to the proteins associated with the ECM as follows: amide I in the range of 1712–1591cm^−1^ (peak 5), the bands for amide II in the range of 1644–1550 cm^−1^ (peak 6), and the bands for amide III in the range of 1300–1200 cm^−1^ (peak 8). Figure 5A shows the FTIR spectrum and the zone of the FTIR spectrum fingerprint (Figure 5B) of hGFs spheroids ranging from 1500 to 400 cm^−1^. In this region, the molecular vibrations are unique for each compound; therefore, this area is important for the characterization of compounds from complex mixtures, such as biological samples. The functional groups identified with the corresponding frequency of the peaks and the vibration modes are described in detail in Table 2.

Taken together, the SEM, EDS, EB, and FTIR analyses provided sufficient evidence to suggest the formation and deposition of spherical microparticles of amorphous calcium phosphate (ACP) on the surface of hGFs spheroids. This finding is interesting because ACP is essential for the formation of mineralized bone and is considered a precursor of apatite crystallization.

### 3.5. Osteoprogenitor Marker Genes Are Significantly Modulated in hGFs-Derived Spheroids NS CL ± P 

After an incubation period of 4 days, hGFs NS CL±P-derived spheroids showed significant modulation for most target genes, as well as high individual variability. The *ALPL* gene decreased significantly for all individuals; in this case, it presented a decrease in expression of an average of 0.05 times (*p* < 0.001) (Figure 6A). The genes that did not show significant changes in expression were *RUNX2*, although expression was decreased in most samples by an average of 0.42 times, and sample No. 5 showed an increase in expression of 9.0 times (*p* > 0.05) (Figure 6B), and the *SPP1* gene exhibited decreased expression by an average of 0.47 times, whereas the expression of sample No. 7 was increased by 3.07 times (Figure 6C).

## 4. Discussion

Alveolar bone grafting is commonly performed to reconstruct the alveolar crest in patients with bone defects associated with CL ± P. The reconstruction of the alveolar process favors permanent tooth eruption, movement of teeth through the alveolar process using orthodontic forces, and re-establishment of esthetics and masticatory function. Bone grafts stabilize the dental arch, optimize the periodontal support of the teeth adjacent to the cleft, and close the oral–nasal clefts, reducing speech difficulties [17,18]; however, bone grafts present limitations for which regenerative medicine attempts to provide a solution.

According to Núñez et al. [19], a limited number of studies have been conducted on cell-based therapy for the regeneration of periodontal tissues; however, hGFs are considered to have the potential to improve the results of regenerative treatment for the reconstruction of alveolar bone and periodontal tissues. In this regard, clinical trials have been carried out using mesenchymal stem cells (MSCs), such as bone marrow stem cells (BMSCs), MSCs derived from dental pulp of extracted teeth (DPSCs), MSCs derived from the periosteum (PdSCs), MSCs derived from the periodontal ligament of extracted teeth (PdlSCs), and MSCs derived from adipose tissue (A-MSCs); however, the complexity, invasiveness, and costs associated with isolating and manipulating these tissues and cells justifies the search for simple, rapid, and inexpensive methods for cell harvesting and processing. The findings of the present study offer evidence of the regenerative potential of hGFs for periodontal bone reconstruction, as well as an easy and accessible cell model for periodontal regenerative medicine procedures.

Current evidence has shown that fibroblasts residing in organs and tissues play an intrinsic proregenerative role by modulating the microenvironment for themselves and/or for MSCs. Although MSCs are considered an ideal source for cell therapy, they still present drawbacks in terms of their clinical application. First, MSCs are scarce cells in adult tissues; therefore, therapeutically relevant doses (1–2 million/kg) must be generated. Consequently, MSCs undergo many cell divisions in vitro, increasing the possibility of mutagenesis and loss of activity. Secondly, implanted MSCs migrate to distant sites of the recipient tissue, suggesting that the mechanism by which MSCs favor tissue regeneration consists in recruiting and/or modulating local cells, which is a potent paracrine effect. Finally, instability in the regenerative potential of MSCs populations has been related to the “stem cell niche” in cell fate, whereby genetic variability and/or epigenetic alterations are involved [20].

One of the most recent approaches for tissue regeneration is the development of 3D culture models derived from stem or somatic cells. However, tissue engineering approaches employing gingival fibroblasts have been poorly studied. Therefore, characterizing hGFs spheroids allows us to extract relevant biological data from this model to select and standardize techniques, methods, and protocols to validate their use in preclinical assays.

Previous studies [21,22] agree that cell markers have not yet been standardized to identify fibroblasts. Various cell types, such as neurons, immune cells, and other non-mesenchymal cells, also express mesenchymal markers that are shared with the fibroblasts. In this study, hGFs in 2D culture expressed Vim, αSMA, Col1A2, Itgβ1, and PDGFRβ and were negative for STIM1 and Stro-1 (Figure 2A). The expression of these markers, along with the morphological characterization of the isolated cells, was sufficient to confirm the phenotype of hGFs. More recently, it was demonstrated that Vim, αSMA, Col1A2, and Itgβ1 are part of the adaptive cellular machinery of the mechanotransduction processes of fibroblasts in response to the physical characteristics of ECM [23]. Similarly, hGFs spheroids in early stages of formation (days 2 to 4 of culture) continued to express the mesenchymal markers Vim, Col1A2, Itgβ1, αSMA, and PDGFRβ (Figure 2B). The presence or absence of Stro-1 has been used to identify subpopulations of MSCs residing in gingival tissue that are characterized by an increased capacity for proliferation and osteogenic differentiation [24]. Our results show that hGFs in 2D and 3D culture were negative for Stro-1, suggesting a phenotype of a differentiated cell niche (Figure 2A).

The liquid overlay technique was used to induce self-assembly of hGFs spheroids. In this study, hGFs spheroids were aggregated and achieved the compaction stage approximately 48 h after cell seeding and reached the consolidation stage on day 4 of culture (Figure 1A). The results show that the average initial diameter of spheroids was 358.90 ± 72.83 µm, and the average diameter on day 10 of culture was 261.44 ± 41.5 µm. This demonstrates that we obtained medium-sized spheroids with the standardization of the described protocol, ensuring the viability of hGFs. Medium-sized spheroids assure the diffusion of oxygen and nutrients while maintaining the integrity of the spheroid. Conversely, when spheroids are larger than 450 µm, the transport by diffusion of oxygen and nutrients to the cells densely grouped in the central region decreases, causing hypoxic centers and leading to necrosis and disintegration of the spheroid [25].

Additionally, all spheroids analyzed in this study showed a gradual decrease in diameter over time (Figure 1B). This is a consistent behavior reported previously in spheroids formed from primary fibroblasts isolated from various healthy tissues [25]. A decrease in spheroid size has been related to the progressive remodeling of the ECM and adaptation of the cell cytoskeleton [26]. Similarly, the reduction in the diameter of primary human dermal fibroblast spheroids, as well as the cessation of proliferation and the adoption of a spherical shape during culture, was described in [25]. In contrast to hGFs and other fibroblast-derived spheroids, human cell lines, such as colorectal carcinoma HT-29 [27], tumor-derived human hepatoma HepG2 [28], and breast cancer cell line MCF-7 [29], that also form spheroids by employing the agarose liquid overlay technique proliferate, generating a gradual and constant increase in spheroid diameter. This behavior correlates with alterations in cell cycle, proliferation, and undifferentiation, which are characteristic of cancer cells. Therefore, the reduction in the size of hGFs spheroids could be related to the downregulation of cell proliferation and the initiation of differentiation [30].

Images generated by SEM show how hGFs spheroids exhibited an increasingly smooth and continuous surface as they matured (Figure 3B). Changes in the appearance of the spheroidal surface have been attributed to high levels of ECM secretion by cells in the peripheral zone of the spheroid [31]. Likewise, the presence of pores on the surface of the spheroids was evidenced, enabling the diffusion of nutrients and oxygen to ensure the viability of the cells in the whole structure. The formation of close cell–cell interactions was also evident (Figure 3B,C). Although we did not determine the nature of these junctions, they modulate multiple signaling cascades, which ultimately determine the morphology and behavior of individual cells and the spheroid as a whole entity.

Furthermore, SEM enabled the identification of the presence of nanoparticles on the surface of the hGFs spheroids, and EDS microanalysis elucidated both semi-qualitative and semi- quantitative information on the deposition or presence of mineral material, mainly from calcium and phosphate, as well as the presence of a crystalline structure by means of EBDS on the surface of the spheroids (Figure 4).

FTIR analysis (Figure 5) enabled the identification of phosphate (PO_4_^3−^) and carbonate (CO_3_^2−^) functional groups on the surface of the hGFs spheroids, with band ranges similar to those of ACP, which is a major transient component of early mineralization in cell culture preceding the formation of apatite in the mature matrix. Furthermore, the results of other molecules, such as lipids and amides of the spheroidal organic phase, corresponded to the peaks and bands reported in the literature (Table 2).

These microanalytical techniques have also been used to study bone matrix calcification during embryonic and postembryonic rat calvaria development and microhydroxyapatite crystal produced by osteoblast cultures [32,33]. These methods are obviously highly specific and sensitive, unlike the non-specific stains that are routinely used to evaluate mineral deposits in cell and tissues, such as alizarin red and von Kossa stain. Therefore, the use of these techniques to evaluate the mineralization processes of cells and tissues in vitro is highlighted.

An interesting result of the immunostaining of hGFs spheroids was the homogenous positive expression of STIM1 throughout the spheroidal surface. STIM1 is a calcium sensor protein localized in the endoplasmic reticulum (ER) membrane that maintains calcium homeostasis by initiating the store-operated Ca^2+^ entry (SOCE) process. SOCE is an important Ca^2+^ influx pathway in non-excitable cells. The widely expressed ER Ca^2+^ sensor protein STIM1 undergoes an intricate activation process in response to Ca^2+^ store depletion and translocates into ER–plasma membrane junctions, where it tethers and activates PM Orai1 Ca^2+^ channels. Ca^2+^ entering through Orai1 channels maintains Ca^2+^ homeostasis [34].

STIM1 is a protein involved in controlling the entry of positively charged calcium ions into cells when ion levels are low, specifically through calcium-release activated calcium (CRAC) channels. By sensing luminal Ca^2+^ concentration, STIM1 protein, in turn, is responsible for relaying the signal of Ca^2+^ store-depletion to pore-forming Orai1 proteins in the plasma membrane. The direct interaction of STIM1 and Orai1 allows for the re-entry of Ca^2+^ from the extracellular space [32]. Moreover, STIM1 and some of the proteins in its interactome have been linked to cytoskeletal reorganization in fibroblasts [34]. Therefore, expression of STIM1 suggests that 3D culture of hGFs induces mechanotransduction pathways in response to changes in the mechanical environment of the niche mediated by store-operated calcium entry (SOCE)/CRAC activation. Furthermore, the fundamental role of Ca^2+^ entry mediated by STIM1/Orai1 during the odontogenic differentiation of MSCs from dental pulp has been demonstrated through the release of exosomes for the mineralization of the ECM [35,36].

The expression of STIM1 in hGFs spheroids is consistent with the results of SEM, EDS, EBSD, and FTIR analyses, which showed the formation of ACP nanoparticles on the surface of the spheroids (Figure 5, Table 1). These mineral deposits increased as the culture time increased; likewise, the Ca/P ratio was variable between the zones of the spheroid. These findings are reported for the first time in this investigation, further suggesting the induction of a mineralizing phenotype of hGFs in 3D culture.

Despite the high variability in the expression of target genes for all individuals, it is important to highlight that for most samples, osteoprogenitor markers showed a significant decrease in expression in 3D culture compared to 2D culture. Gene *ALPL* presented an average of 95% decrease in expression for all samples without exception (Figure 6A). This result is most likely related to the mineralization process of the spheroid surface, suggesting osteogenic differentiation of hGFs NS CL ± P according to the research carried out by Yamamoto et al. [37] in immortalized mouse dental papilla cells. In that case, the cells were cultured in low-adherence 96-well plates without any exogenous factor inducing differentiation. Similar to this investigation, a low expression of *ALPL* was observed on day 1 of culture. However, in the immortalized cells of the dental papilla, the expression of *ALPL* increased during the 14 days of culture, as well as the rest of the osteogenic markers that were evaluated by RT-qPCR, evidencing the formation of mineralized nodules with red alizarin on day 2 of culture, which increased over time. Additionally, the change in the expression of *RUNX2* and *SPP1* was not statistically significant. These data indicate that on day 4 of culture of the spheroids, the expression of the *RUNX2* and *SPP1* genes remained similar to the that in the monolayer culture, that is, the 3D culture failed to modulate their expression (Figure 6B,C). It is important to note that cell behavior and differentiation depend on a complex network of signals that operate in different periods and time scales; therefore, the modulation of osteoprogenitor markers in hGFs over time in 3D culture requires further evaluation.

In addition, the most recent research on osteogenic differentiation of human MSCs in 3D culture, with or without the addition of external factors, such as differentiation media or induction molecules, shows some similarities with the 3D culture of hGFs reported here. For example, cell condensation that occurs in 3D culture is equal to or even more important than chemical induction for cells to undergo osteogenic differentiation by recapitulating the mesenchymal condensation process during ossification. Furthermore, hypoxic conditions and actin depolymerization were reported to favor the cell differentiation process [38,39]. Similar processes occur during 3D culture of hGFs. Lastly, the ultrastructure of the hGFs spheroids is similar to that of the surface of the spheroids of an osteoblast cell line generated using the magnetic levitation technique. In this case, SEM showed close contact between cells and the formation of a three-dimensional network of small structures, indicating the deposition of a mineralized ECM [40].

The mechanism by which hGFs spheroids deposit minerals in 3D culture conditions remains to be determined. However, according to Monterubbianes, hGFs may undergo a partial osteogenic differentiation process [41], or hGFs may be better equipped to respond to a mechanical type of stimulation than chemical induction.

## 5. Conclusions

Within the limitations of the present study, we can conclude that hGFs spheroids acquire a mineralizing phenotype as a result of 3D cytoskeleton reorganization and the possible activation of ion channels, which could ultimately translate into changes in Ca^2+^ hemostasis and SOCE/CRAC activation. Additional studies are necessary to identify the role of STIM1, which could mediate the processes of amorphous calcium phosphate nanoparticle deposition on the hGFs spheroidal surface and the mineralizing environment in the niche, as well as its possible implication in periodontal regeneration and translational dentistry.

## Figures and Tables

**Figure 1 cells-11-03647-f001:**
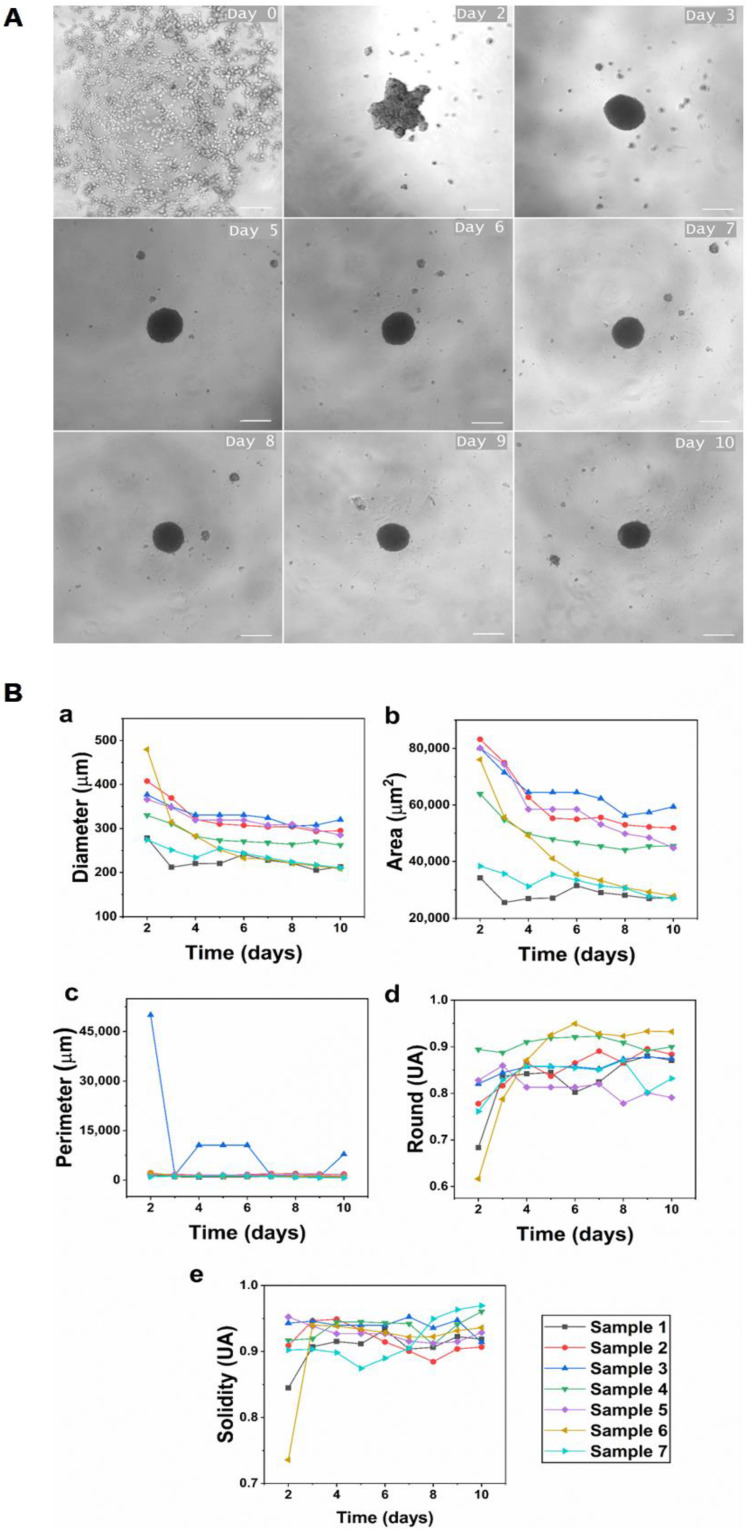
Dynamics of hGFs spheroid formation. (**A**) Spheroid formation of human gingival fibroblasts. Day 0: seeding of 5 × 10^3^ cells in the cell suspension well; 24 to 48 h: cell aggregation stage; day 3: formation of a dense cellular spheroid; days 4 to 5: spheroidal compaction stage; days 6 to 10: formation of a mature spheroid. Scale bar, 200 µm; 5× magnification. (**B**) Morphological changes of spheroids from day 2 to 10 of culture: (**a**) diameter, (**b)** area, (**c**) perimeter (**d**) roundness, and (**e**) solidity.

**Figure 2 cells-11-03647-f002:**
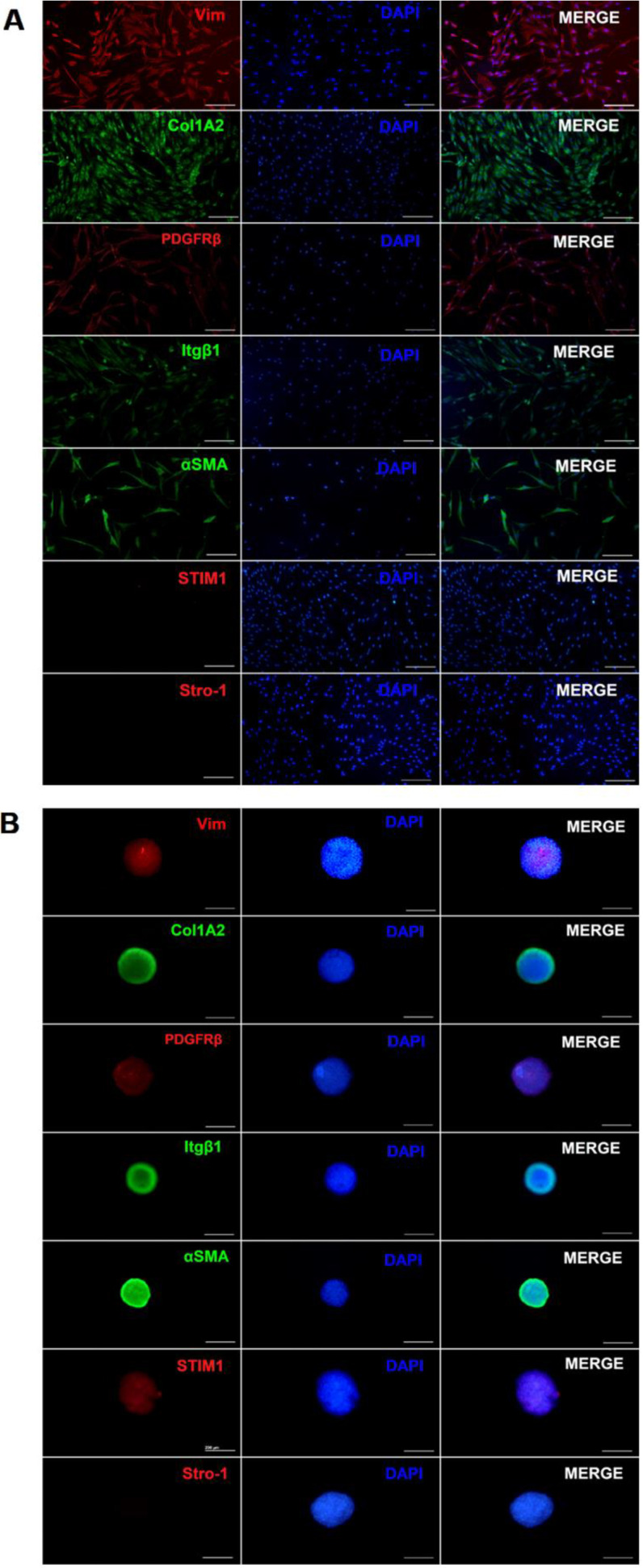
Immunofluorescence analysis. (**A**) Characterization by of a population of hGFs. Positive markers panel: Vim, Col1A2, PDGFRβ, Itgβ1, and αSMA. Cells were negative for STIM1 and Stro-1. (**B**) hGFs spheroids express STIM1. Scale bar, 200 µm; 10× magnification. The data are from three representative experiments.

**Figure 3 cells-11-03647-f003:**
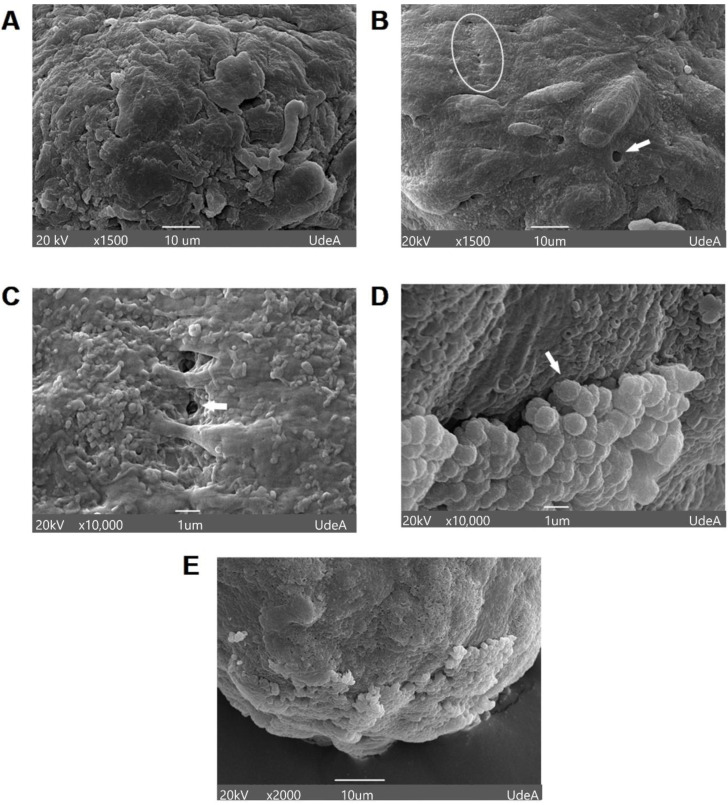
Morphological and topographic analysis of the surface of hGFs spheroids. (**A**) Scanning electron microscope microphotographs of a young spheroid after 3 days of culture with a round but irregular surface. (**B**,**C**) The presence of close cell–cell interactions (white circle) and multiple pores in a mature spheroid after 9 days of culture (white arrow). (**D**) The presence of multiple spherical nanoparticles on the surface of the spheroids is indicated by the white arrow (9 days of culture). (**E**) Higher magnification shows spherical nanoparticles with a rough appearance.

**Figure 4 cells-11-03647-f004:**
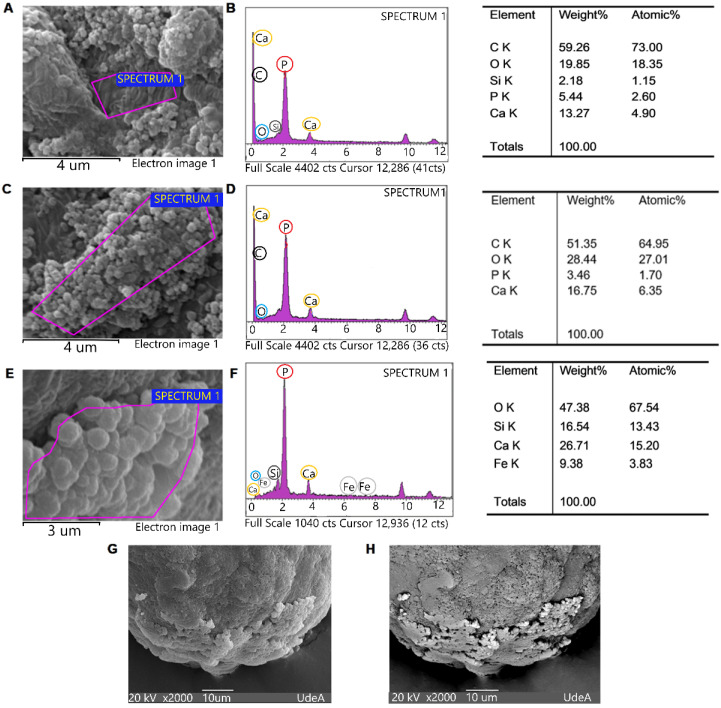
Ultrastructural analysis of the spheroidal surface of hGFs. (**A**,**C**,**E**) SEM images with the corresponding EDS spectrum and tables. (**B**,**D**,**F**) Detected ions (carbon (C), oxygen (O), phosphorus (P), calcium (Ca), and silica (Si)) in the purple area delimited in the SEM image of a spheroid. (**A**,**B**) Spheroid after 3 days of culture. (**C**,**D**). Spheroid after 21 days of culture. (**E**,**F**) Spheroid after 9 days of culture. (**G**) SEM and (**H**) EB images of a 9-day culture spheroid at lower magnification. Scanning electron microscope (SEM), spectrum of energy dispersive X-ray spectroscopy (EDS), and backscattered electrons (EB).

**Figure 5 cells-11-03647-f005:**
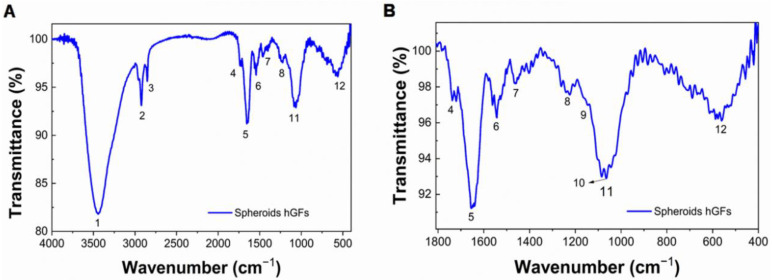
FTIR spectrum of hGFs spheroids. (**A**) Fourier transform infrared (FTIR) spectrum of a group of hGFs spheroid after 21 days of culture. (**B**) The range of 400–1600 cm^−1^ corresponds to the zone of the FTIR spectrum fingerprint. Peak positions represent molecular vibrations characteristic of mineral and organic molecular vibrations. The distribution of the mineral components of hGFs spheroids showed v1–v3 [CO_3_^2−^] (peak 7) at~1423 and 1404 cm^−1^, v1–v3 [PO_4_^2−^] (peak 11) at~1064 cm^−1^, and v4 [PO_4_^3−^] (peak 12) at~557 cm^−1^, and organic molecular vibrations (Sugar ring stretch C-O, peak 10) corresponds to carbohydrate-proteoglycan functional groups (arrow).

**Figure 6 cells-11-03647-f006:**
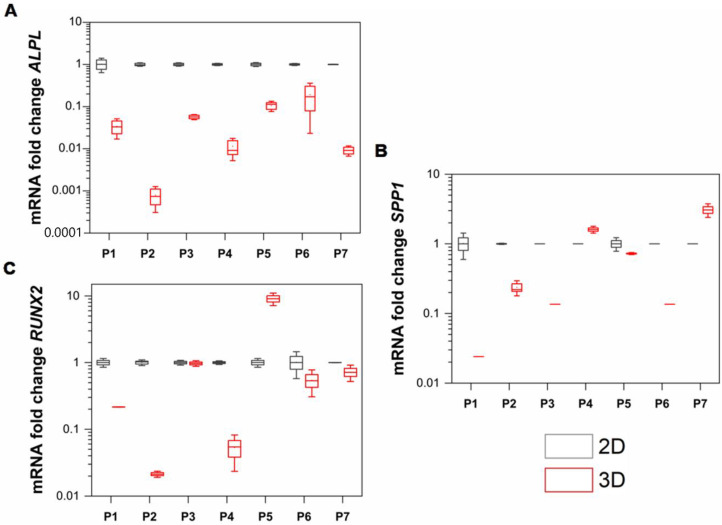
(**A**) *ALPL* expression (*p* < 0.001). (**B**) *RUNX2* expression (*p* > 0.05). (**C**) *SPP1* expression (*p* > 0.05).

**Table 1 cells-11-03647-t001:** Primer pairs used for RT-qPCR.

Gene	Primer Sequence	Product Size (bp)	Annealing Temp (°C)	Gene Reference
*ALPL*	F: CTATCCTGGCTCCGTGCTCC R: TTAACTGATGTTCCAATCCTGCG	82	57	NM_000478.6
*RUNX2*	F: GACCAGTCTTACCCCTCCTACC R: CTGCCTGGCTCTTCTTACTGAG	190	60	NM_001024630.3
*SPP1*	F: CAACAAATACCCAGATGCTGTGGC R: GGACTTACTTGGAAGGGTCTGTGG	94	57	NM_000582.3
*GAPDH*	F: CCTGCACCACCAACTGCTTA R: GGCCATCCACAGTCTTCTGAG	120	60	NM_001357943.2
*RPL27*	F: TGAAACCTGGGAAGGTGGTGC R: TCTTGGCGATCTTCTTCTTGCC	180	60	NM_001349921.2

**Table 2 cells-11-03647-t002:** Relevant peaks of the FTIR spectra of hGFs spheroids.

Peak Number	Frequency (cm^−1^)	Functional Groups	Vibrational Mode	Band/Peak Reference (cm^−1^)	Reference
1	3444.86	OH	Water bound to samples		
2	2956.87	CH_3_ Methylene groups of cellular lipids	Asymmetric stretch mode CH2	Lipids (3000–2700) 2946–2889	[12]
3	2852.71	Saturated alkyl chains of cellular lipids	=CH Stretch mode	2990–2836	[12]
4	1737.86	COO- Carboxylic ester of fatty acids	C=O Stretching of the ester fraction	1740	[12] [13]
5	1649.13	Amide I	C=O Stretching	Proteins (1700–1500) 1650 Overlapping peak (CO_3_)	[14] [15]
6	1544.98	Amide II	C-N stretching and N-H flexion	1540–1550	[13] [14]
7	1460.11 1423 and 1404	CO_3_^2−^	v_3_ CO_3_	Remarkable band with peaks around 1550–1414 cm^−1^	[14] [15] [16]
			v_2_ CO_3_	Remarkable band around 875 cm^−1^	[14]
8	1224.79	Amide III		1240	[14]
9	1143	HPO_4_^2−^	v_3_ HPO_4_^2−^	1143, 1163	[15]
10	1064	Carbohydrates-proteoglycans	Sugar ring stretch C-O	1060 Overlapping peak	[13] [15]
11 12	1192–935 1064 557	PO_4_^3−^	v_1_–v_3_ PO_4_^2−^	Broad band with a maximum around 1030 cm^−1^ and a discreet shoulder around 1095 cm^−1^	[13] [14] [15] [16]
			v_4_ PO_4_^3−^	Partially at two distinct peaks around 604 and 563 cm^−1^ 569, 590	[16]

## Data Availability

Not applicable.

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
