# Peer review of "Ultrastructural Characterization of Human Gingival Fibroblasts in 3D Culture"

_cells, 2022, doi:10.3390/cells11223647_

Round 1

Reviewer 1 Report

Dear authors

Can you explain why you chose agarose to make spheroids? Would other techniques give the same results?

Can you detail the immunofluorescence protocol on the spheroids: is it on sections or on the whole spheroids?

Show immunofluorescence for all markers in spheroids. I am not convinced by the staining of Stim1. Can you improve the images?

To show the junctions, you could do TEM and immunofluorescence with the junction proteins like ZO1, occludin, connexin...

Have you observed signs of necrosis or apoptosis in the spheroids?

Regarding figure 5, the C is missing.

How to explain the presence of nanoparticles on the surface of spheroids?

Best regards

Reviewer 2 Report

The paper entitiled "Ultrastructural characterization of human gingival fibroblasts 2 in 3D culture" clearly describes a preclinical 3D culture model of primary human gingival fibroblasts and the fact that these 3D organoids  are able to generate  a sort of mineralized environment in culture. 

Nevertheless I have some concerns to ameliorate the paper: 

- have you tried to investigate the presence of the CD90 or CD105 markers? They are extremely present in mesenchymal stem cell cultures;

- could you show the presence/absence of some genes or proteins related to osteogenesis? Like Osteocalcin or osteopontin. 

- what is the advantage of organoids created with the liquid overlay technique rather than 3D printing? Are you sure that you are able to create the in vitro environment for bone maturation?

- can you explain please the pathway of Stim 1?

Reviewer 3 Report

The evaluated manuscript entitled "Ultrastructural characterization of human gingival fibroblasts in 3D culture" is a well-written article. The authors, based on the results obtained, suggest that human gingival fibroblasts have an intrinsic potential to generate a mineralized niche in 3D culture. This indicates its possible implication in periodontal regeneration as well as in translational dentistry.

In line number 264 the figure number should be corrected (is: "Figure 1", should be: "Figure 2").

In my opinion the figure 1B and figure 2A is too small and illegible.

Round 2

Reviewer 2 Report

Thank you very much for answering me in a clear and thorough way.

The paper is ready for publication.